

# A novel trace-based sampling method for conformance checking

Heidy M. Marin-Castro[1], Miguel Morales-Sandoval[2],
José Luis González-Compean[3] and Julio Hernandez[1]

[1] Universidad de las Américas, Cholula, Puebla, Mexico
[2] Computer Science, Instituto Nacional de Astrofísica, Óptica y Electrónica, Tonantzintla, Puebla, Mexico
[3] Cinvestav Tamaulipas, Ciudad Victoria, Tamaulipas, Mexico

## ABSTRACT

It is crucial for organizations to ensure that their business processes are executed accurately and comply with internal policies and requirements. Process mining is a discipline of data science that exploits business process execution data to analyze and improve business processes. It provides a data-driven approach to understanding how processes actually work in practice. Conformance checking is one of the three most relevant process mining tasks. It consists of determining the degree of correspondence or deviation between the expected (or modeled) behavior of a process *vs* the real one observed and revealed from the historical events recorded in an event log during the execution of each instance of the process. Under a big data scenario, traditional conformance checking methods struggle to analyzing the instances or traces in large event logs, increasing the associated computational cost. In this article, we study and address the conformance-checking task supported by a traces selection approach that uses representative sample data of the event log and thus reduces the processing time and computational cost without losing confidence in the obtained conformance value. As main contributions, we present a novel conformance checking method that (i) takes into account the data dispersion that exists in the event log data using a statistic measure, (ii) determines the size of the representative sample of the event log for the conformance checking task, and (iii) establishes selection criteria of traces based on the dispersion level. The method was validated and evaluated using fitness, precision, generalization, and processing time metrics by experiments on three actual event logs in the health domain and two synthetic event logs. The experimental evaluation and results revealed the effectiveness of our method in coping with the problem of conformance between a process model and its corresponding large event log.

## INTRODUCTION

Process mining (PM) is a discipline that aims to understand the actual behavior of business processes better and derive related information, such as performance, by extracting knowledge from events recorded during the execution of the process (*Sypsas & Kalles, 2022*). Each of these executions represents an instance or case of the process, and the

Corresponding author
Miguel Morales-Sandoval,
mmorales@inaoep.mx

related information for each case expressed as a set of events in a trace is commonly stored in an event log.

Typically, business processes are first designed and documented by experts. Later, business processes are implemented and executed. Due to digitalization, business processes are more commonly supported by information systems. While a documented process only provides an idealized view of how the process should be executed, the event log can reveal the real behavior of the process, usually not the same as the expected one. To maintain the expected behavior of a business process as documented, it is necessary to know and analyze the discrepancies or deviations that may exist according to the actual behavior of a business process. There is a variety of situations in which deviations of business process in execution can occur (*Carmona et al., 2018*), such as:

–Wrong recording of activity executions.
–Interruption of the process execution.
–Corruption in the recorded event data.
–Technical problems with the information systems.
–Poor data quality (missing, erroneous or noisy values, duplicates, *etc*.).
–Synchronization problems.
–Decisions taken that violate some organizational internal rules or external regulations.
–Lack of coordination among the actors involved in the process.

In the health domain, business processes present high levels of data dispersion and variation given the nature of clinical business processes related to the diagnosis, treatment or monitoring of patients and the organization of health workers. Thus, health processes are constantly evolving, and their activities are changing dynamically, which makes it challenging to preserve the original flow of the process. Through PM, it is possible to determine the value or level of deviation (or correspondence) between the activities registered in the event log (execution) with the activities in the idealized process (documented). This is the objective of the conformance checking task in PM, which aims at identifying if the activities of the process are correctly followed by the actors in the process and hence, to identify a problem or undesired behavior in the process. Through conformance checking, it is possible to identify bottlenecks, anticipate problems, record policy violations, recommend counter-measures, and in general, improve the performance of the process (*van der Aalst, 2022*; *Polyvyanyy, Moffat & García-Bañuelos, 2020*; *Imran, Ismail & Hamid, 2023*).

Conformance checking is highly desirable in environments such as healthcare (*Benevento et al., 2023*), making it possible to resolve possible issues that translate into more efficient, high-quality processes that adhere to clinical guidelines while lowering operational costs. If considering that the vast majority of health processes have a high degree of freedom in the execution of activities which contribute to the construction of alternative paths in the execution of the process, then not only effective but also efficient conformance-checking techniques are required to work with large event logs.

Nowadays, many approaches have been proposed in the literature for the conformance-checking task, being one of the most used replay techniques. It relates and compares the behavior expected in the process model with the behavior observed in its event log by replaying each trace of the log in the process model. Different replay techniques exist, alignment (*Bose & van der Aalst, 2010*; *de Leoni, Maggi & van der Aalst, 2012*; *de Leoni & Marrella, 2017*) being one of the best-known approaches. The challenge of such techniques is to replay most of the log traces in the process model, either documented or discovered. The higher the number of traces replayed, the better the conformance metric. However, the conformance metric can be significantly degraded if the process is highly flexible at execution time and many of the previously-mentioned deviation situations occur. If this is the case, the traces in the event log may have high variability and hence low success during the replaying attempt. Furthermore, most of the known conformance techniques are very time-consuming. That is why, with the explosive growth of big data, conformance techniques cope with huge event logs, which may make the conformance task intractable for real-life scenarios (*Rozinat & van der Aalst, 2008*; *Bose & van der Aalst, 2012*; *Munoz-Gama, Carmona & van der Aalst, 2014*; *Grigore et al., 2024*). This problem motivates the design of efficient conformance-checking techniques in the context of PM in big data scenarios.

This article presents a novel conformance-checking method well suited for big data scenarios. As distinctive, it uses an instance selection approach based on an analysis of data dispersion in the event log and an approximation algorithm for conformance calculation. The proposed method aims at finding a value close to the exact conformity solution (as if the entire event log was used), but just considering a representative traces samples subset of the event log. This way, the goal is to reduce the computational workload and hence the execution time of the conformance task with the most negligible impact on fitness. The trace selection mechanism obtains a representative number of traces in the event log using a statistical measure to establish the approximate but effective amount of traces that should be considered in the subset without deviating from the original behavior implied by the original data set. In summary, the main contributions of this work are the following:

–A novel data dispersion metric applied in the PM context, particularly in event log data for the conformance checking task.

–A novel conformance checking method using a traces selection mechanism and an approximation algorithm for computing the conformance value.

Our proposed sampling method succeeds in identifying subsets of trace samples that represent the event logs' behavior, reducing the processing time in the conformance task by up to 90%.

The rest of this article is organized as follows. "Preliminaries" presents background on PM and conformance checking. "Novel Conformance Checking Method" describes our proposed approach for conformance checking and the instance selection strategy. "Experimental Evaluation and Results" presents the experimental settings, results, and

discussion. "Related Work and Comparisons" presents a review of related works. Finally, "Conclusions" concludes this work and sketches future work directions.

## PRELIMINARIES

This section summarizes some of the essential and relevant concepts for designing our proposed conformance-checking method. In particular, in this section we present the basis for PM and conformance checking.

### Process mining

PM is a research area that intersects data science and process science techniques to perform model discovery, conformance checking, and enhancement tasks of business processes by extracting knowledge from event logs. It is focused on studying and improving operational processes, *i.e.*, processes requiring the repeated execution of activities to deliver products or services (*van der Aalst, 2022*). PM is supported by two main components: business process models and event logs.

   A business process model is used to visualize and understand the behavior of a business process. It can be represented in various notation languages (Petri Net, BPMN, workflow, *etc*). A BPMN model of a business process can be formally represented as a directed connected graph $M = (N, E_m)$, where $N$ corresponds to a set of nodes and $E_m$ is the set of directed edges that connect those nodes. $N$ represents various nodes in the graph, generally associated with tasks in the business process or elements to control the flow of tasks according to the business process logic. $N$ includes nodes in the tuple $\{i, o, T, G^+, G^*, G^o\}$, where $i$ is the node representing the initial event or starting activity in the process, $o$ is the end event or ending activity in the process, $T$ is the set of all the tasks executed in the process. $G^+$ is the set of nodes representing AND gates. The outgoing edges of an AND-gate node point to tasks that must be executed in parallel. $G^*$ is the set of nodes representing XOR gates. Just one of the tasks pointed by the outgoing edges of an XOR-gate node is executed (depending on a given condition). $G^o$ is the set of nodes representing OR gates. Any or even more than one task pointed by the outgoing edges of an OR-gate node could be executed. An arc in $M$ of the form $(a; b)$ implies that if $a, b \in N$, $b$ is a task executed after the execution of $a$. Figure 1 shows an example of a process model in health using the BPMN notation previously described. The process begins with the activity *Register*, followed by the parallel activities *Collect symptoms*, *Triage*, and *Record vital signs* (tasks executed in any order), followed by the activity *Update medical record* and the activities *Take blood test* and *Check circulation*, where only one of these last two activities can be executed. In the process model, the initial event (depicted by the circle to the left of Fig. 1) is triggered by the patient upon arrival. It ends with a final event (depicted by the circle to the right of Fig. 1). During the execution of the healthcare process, different actors are involved, such as healthcare personnel (physicians, nurses, assistants, *etc*.) and patients performing one or more activities. Business process models can be created by experts (documented model) or discovered from the event log (discovered model). Process model discovery is the most popular task in PM.

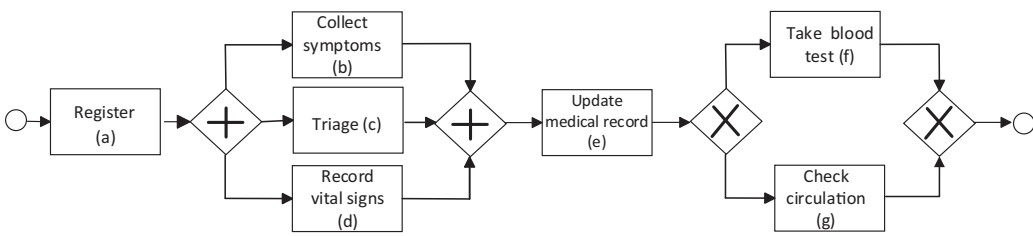

**Figure 1 Example of a BPMN process model in the health domain.**

**Table 1 Event log example in the health domain.**

| Case ID | Event ID | Timestamp | Activity | Cost | Resource | Patient |
|---|---|---|---|---|---|---|
| 1100 | 16 | 25-03-2012:10.00 | Register | 100 | Assistant | Mike |
| 1100 | 17 | 25-03-2012:11.08 | Triage | 50 | Nurse | Mike |
| 1100 | 18 | 25-03-2012:11.28 | Collect symptoms | 100 | Nurse | Mike |
| 1100 | 19 | 25-03-2012:13.15 | Update record | 50 | Physician | Mike |
| 2200 | 19 | 25-03-2012:13.30 | Register | 100 | Assistant | Robert |
| 2200 | 20 | 25-03-2012:16.00 | Triage | 500 | Nurse | Robert |
| 2200 | 21 | 25-03-2012:17.30 | Update record | 800 | Nurse | Mike |
| 3300 | 22 | 02-04-2012:17.50 | Register | 100 | Assistant | Susan |
| 3300 | 23 | 02-04-2012:19.20 | Collect symptoms | 50 | Physician | Susan |

Event logs are created from the data collected during the execution of business processes activities, in the context of process instances (or cases). The execution of an activity in a given process case is referred to as an event $e_j$. The sequence of all the events for a specific process case $i$ is also referred to as a trace $t_i = e_1, e_2, e_3, \ldots, e_m$. An event is associated with an activity in the business process. Thus, multiple traces can contain the same sub-sequence of activities, yet, since events are unique, each trace itself contains different events. Therefore, activities (associated to events) may appear more than once in a trace and in the entire log (*Marin-Castro & Tello, 2021*). Let $A$ be the set of activities in the business process and $M(A)$ a multiset over $A$, where elements of $A$ can appear multiple times. Formally, an event log $L$ can be defined as a multiset of sequences over $A$, *i.e.*, L $\in M(A^*)$. Table 1 shows an example of an event log associated with the implementation and execution of the documented process model in Fig. 1. In this example, three cases of the process are shown (three traces) {<*Register*, *Triage*, *Collect symptoms*, *Update record*>, <*Register*, *Triage*, *Update record*>, <*Register*, *Collect symptoms*>}. In the example, activities associated to the events are shown in each trace.

## Conformance checking

The goal of conformance checking is to relate the modeled behavior of a process, either in a documented or discovered model, to the recorded one in an event log. The general idea is to compare and analyze observed instances of a process in the presence of a model, either discovered or manually constructed (*Carmona et al., 2018*).

Conformance checking allows knowing some of the following important aspects of the business process:

–Is the process being executed as it is documented in a model?

–Is the model of a process still up-to-date?

–Is compliance with standards and regulations maintained in the execution of process activities?

–What is the level of flexibility allowed in the execution of the process?

Conformance checking may expose undesirable deviations in a business process and, thus, guide optimizations and improvements for the process. However, deviations could be desirable and intentionally generated by actors in the process to improve execution paths, not formerly foreseen at design time. Thus, while conformance-checking is a useful tool for decision-making, alignment is one of the most common conformance-checking techniques to achieve that goal.

Alignment is defined as a sequence of moves, each relating an event in the trace to an activity in the model. According to the cost function, an alignment with the lowest cost is optimal. Unfortunately, the problem of finding the optimal alignment is NP-hard (*de Leoni & van der Aalst, 2013*). This means that when the process is large, it is intractable to determine the optimal alignment, or if the event log is large, it may not be possible to reach or obtain the conformance value. That is why the motivation of this work is to propose an alternative method that addresses the alignment problem using a representative subset of the event log to estimate an approximate but confident conformance value.

In an alignment technique, an individual mapping of an event $e_j$ (in a trace $t_i \in L$) to an activity $a_k$ (in a model $M$) is referred to as *a move* and denoted by $\varepsilon$. There are four types of moves:

–*Synchronous move* (the move in both the log and the model): $e_j$ in any $t_i$ trace can be mapped to the occurrence of an enabled activity $a_k$ in $M$. This move can be expressed as the mapping $(e_j, a_k)$.

–*Move on model* ($move_M$): The occurrence of an enabled activity $a_k$ in $M$ cannot be mapped to any event given the flow implied by a trace $t_i$. In this case, the behavior in $M$ would not be observed in the trace. This move is denoted as a mapping $(\gg, a_k)$. Here, the symbol $\gg$ denotes the absence of event $e_j$ that should correspond to $a_k$ in $M$.

–*Move on log* ($move_L$): Opposite to the previous case, an event $e_j$ in a trace $t_i$ cannot be mapped to any enabled activity in $M$. This move is expressed by the mapping $(e_i, \gg)$. Now, $\gg$ denotes the absence of activity $a_k$ in $M$ that should correspond to $e_j$ in $t_i$.

–*Illegal move* ($move_I$): This move occurs when $e_j$ belonging to $t_i$ cannot be mapped in any way to an activity $a_k$ in $M$.

As an example, consider Fig. 2 showing three possible alignments of the trace $<a, b, c, e, f>$ in the BPMN process model $M$ in Fig. 1. In the three alignments, the mapping $(a, a)$ is a synchronous move and refers to the first event in the trace corresponding to the

$\gamma_1=$

| a | b | c | >> | e | f | >> |
|---|---|---|----|---|----|----|
| a | b | c | d  | e | >> | g  |

$\gamma_2=$

| a | >> | b  | c | >> | e | f |
|---|----|----|---|----|---|---|
| a | d  | >> | c | b  | e | f |

$\gamma_3=$

| a | b | c | >> | e | >> | f  |
|---|---|---|----|---|----|----|
| a | b | c | d  | e | g  | >> |

**Figure 2 Possible alignments of trace $<a, b, c, e, f>$ for the model in Fig. 1.**

first internal activity in the model $M$. While the second move in $\gamma_1$ and $\gamma_3$ is a synchronous move, it is a log move in $\gamma_2$. In the example, each alignment has the same number of move types, but not the same moves: four synchronous moves, two log moves, and one model move. As shown in the previous example, many possible alignments may exist between a process model and an event log. Let $A_{LM}$ be the set of legal moves implied by the alignment technique over the model $M$ associated with the set of traces $\in L$. An alignment $\gamma$ is a subset $\{\varepsilon_1, \varepsilon_2, \ldots, \varepsilon_k\}$ of $A_{LM}$, and corresponds to the alignment for a specific trace in $L$, for example, any of the ones presented in Fig. 2.

Let $\delta$ be the cost function $\delta: A_{LM} \rightarrow \mathbb{N}$ that assigns costs to legal moves in $A_{LM}$. Thus, the cost of an alignment $\gamma \in A_{LM}$ can be computed as $C_\gamma = \sum_{i=1}^{k} \delta(\varepsilon_i), \varepsilon_i \in \gamma$ (*Munoz-Gama, 2014*). The costs may depend on the importance given to the activities within the process, *i.e.*, some activities may affect the execution of the process more than others, so their omission or change of order during execution may cause severe problems. Conversely, other activities are more tolerable. Moreover, the severity assumed for log and model moves may be different, depending on the level of affectation. A standard cost function $\delta_S$ assigns costs to moves as follows:

–A synchronous move has cost 0: $\delta_S(e_j, a_k) = 0$.
–A log move has cost 1: $\delta_S(e_j, >>) = 1$.
–A model move from a visible task has cost 1: $\delta_S(>>, a_k) = 1$.
–A model move from an invisible task (task without label) has cost 0: $\delta_S(>>, a_k) = 0$.
–An illegal move always has cost 0.

Consider $\delta_S$ for the alignments in Fig. 2. For example, for $\gamma_1$, its cost is $\delta_S(\gamma_1) = \delta_S((a, a)) + \delta_S((b, b)) + \delta_S((c, c)) + \delta_S((>>, d)) + \delta_S((e, e)) + \delta_S((f, >>)) + \delta_S((>>, g)) = 0 + 0 + 0 + 1 + 0 + 1 + 1 = 3$. Since the move types are the same in the three alignments, the cost will also be the same in the three cases.

## NOVEL CONFORMANCE CHECKING METHOD

This section presents the proposed conformance-checking method based on a traces selection approach. The key element in this method is obtaining a subset of representative

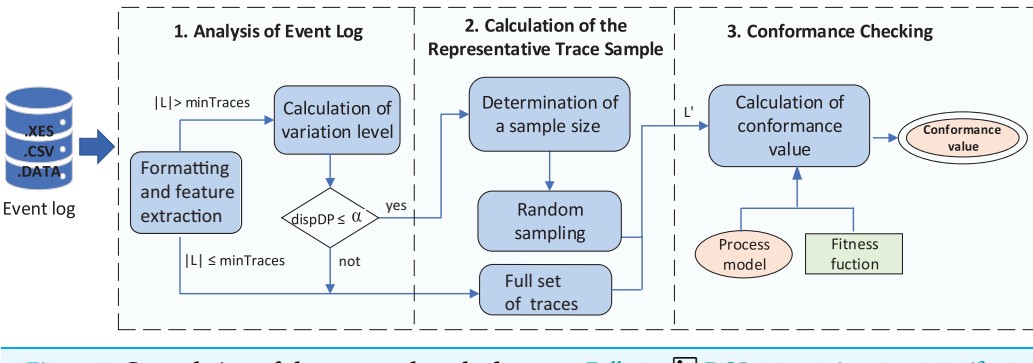

**Figure 3  General view of the proposed method.**     

traces of the complete collection of events, thus allowing the discovery of an approximate but confident value of conformance.

The proposed method consists of the following three stages (shown in Fig. 3): (1) analysis of the event log; (2) sample selection of representative traces; and (3) approximate conformance checking computation. The entire flow transforms and analyzes an event log given as input. Based on the number of traces in the event log and their dispersion level, a sample of traces (representative of the entire event log) is selected and used to compute the approximate conformance metric between the model and the event log. The following sections describe in more detail the method's stages.

## Analysis of the event log

In the first stage, the event log is received as input for further processing to extract its features such as #events, #activities, and data associated to traces. The cardinality $|L|$ is used to determine how to proceed with the log to compute the sample of traces in the next stage. In case $|L|$ is greater than a given threshold ($minTraces$), it could be worthwhile to invest some effort in selecting an instance subset of the event log to reduce the workload in the conformance checking task. If this scenario holds true, we calculate the dispersion level ($dispDP$) of $L$ to assess the feasibility of employing a representative trace sample for the fitness computation. If the dispersion level does not exceed a specified threshold ($\alpha$), we proceed to obtain the sample of traces in Stage 2.

Stage 1 and stage 2 of Fig. 3 are reflected in Algorithm 1. The data structure that models the event log $L$ allows obtaining associated data such as the names of activities in the event log, total number of events, size of a trace $t_i \in L$, frequency of occurrence of a given activity in a trace $a_i$ or in all the event log, total number of activities in the event log, among others.

In order for our method to be applied, it is checked that the cardinality of the input event log is greater or equal to $minTraces$. This threshold is defined at implementation time. Usually, event logs of size average to large have an amount of cases or traces greater than $minTraces$ to select a sample of traces for the conformance check task. Therefore, if a log with the described characteristics is received, the level of dispersion $dispDP$ of traces in $L$ is calculated (lines 4–20 in Algorithm 1). We define $dispDP$ as a metric that allows measuring the variability or spread of traces (sequences of events) within an event log. An event log with low dispersion and a large number of traces suggests uniformity among

**Algorithm 1** Computation of sample subset L′, representative of the complete log L (Stages 1 and 2 of Fig. 3).

**Data:** *L*: event log, *minTraces*: predefined size, α: dispersion threshold

**Result:** *L′*: A traces sample of *L*

1  $n \leftarrow |L|$;

2  **if** *(n > minTraces)* **then**

3      $Activ[] \leftarrow activities \in L$;

4      **for** $t_i \in L$ **do**

5          $s \leftarrow t_i.length$;

6          $S_i = \frac{s}{events}$;

7          **for** $a_j \in Activ$ **do**

8              $V_{j,i} \leftarrow 0$;

9              **if** $(a_j \in t_i)$ **then**

10                  $V_{j,i} \leftarrow \frac{t_i.getFrequencyOfActivity(a_j)}{L.totalFrequency(a_j)}$;

11              **end**

12          **end**

13      **end**

14      **for** each $S_i$ **do**

15          **for** $a_j \in Activ$ **do**

16              $dispDPActiv_j \leftarrow 0.5 \times \sum_{k=1}^{n} |V_{j,k} - S_i|$;

17              $dispDPnorm_j \leftarrow \frac{dispDPActiv_j}{1 - \frac{1}{n}}$;

18          **end**

19      **end**

20      $dispDP \leftarrow \frac{\sum_{j=1}^{|Activ|} dispDPnorm_j}{|Activ|}$;

21      **if** $(dispDP \leq \alpha)$ **then**

22          $nSample \leftarrow$ `determineSizeOfSample` $(L)$;

23          $L' \leftarrow$ `randomSampling` $(nSample, L)$;

24          **return** $L'$;

25      **end**

26  **end**

27  $L' \leftarrow L$;

28  **return** $L'$;

traces, allowing for simpler trace analysis through a sampling. Conversely, an event log with high dispersion has a wide variety of execution instances whose log behaviour may not be retrieved by simple trace sampling. When dealing with an event log with low dispersion, a subset of traces reduces the computational load, enabling faster processing while still yielding meaningful insights about the overall process. In the context of process mining and for this work, a data dispersion measure was used to identify the distribution of

events in the traces. This was done because the content or characteristics of the event logs are not known in advance. Dispersion is a statistical measure widely used in natural language processing for *corpus* linguistics. In this work, we are using the dispersion measure for the first time in the event log analysis task of the proposed method. For this, a series of basic statistical measures, their characteristics and their manual calculation were studied. Derived from this study, the normalized version of the deviation of proportions (DP) dispersion measure (*Gries, 2020*) was selected and used. The DP dispersion metric can handle differently sized event logs, and it can immediately be applied to other kinds of data/scenarios such as activities co-occurrence frequencies in event logs. The DP measure can distinguish distributions that other measures fail to distinguish, measures do not have too high sensitivity unlike some other metrics, and it does not output extremely high values when low expected frequencies come into play.

DP for a given activity $j$ in a trace $i$ is calculated according to Eq. (1) as a value in the range [0,1]. A value closer to 1 indicates a higher dispersion in the event log. In that equation, $f$ is the overall frequency of an event in the event log; $V_{j,k}$ (computed at line 10 in Algorithm 1) represents the relation of the frequency of activity $j$ in the trace $k$; and $S_i$ (computed at line 6 in Algorithm 1) represents the relation of total events in the trace $t_i$ to the total events in the log.

$$DPActiv_j = 0.5 \times \sum_{k=1}^{n} \left| V_{j,k} - S_i \right|. \tag{1}$$

## Representative sample of traces

In this stage, we compute a representative sample (traceSample) from the complete set of traces. This sample serves as the input traces set for the conformance checking task in stage 3. Once the alignment value of a trace is determined, no additional time is needed to calculate the alignment of its similar traces. However, our approach aims to capture the behavior of the entire event log, especially focusing on traces with lower similarity.

According with our method, the sample of traces could be obtained in one of two forms: (1) by random sampling or (2) by selecting the full set of traces. The first case is when $(n = |L|) > minTraces$ and $dispDP \leq \alpha$. The second one is when $(n \leq minTraces)$ or $dispDP > \alpha$.

### *Sample traces by random sampling*

This case is implemented in lines 21–24 in Algorithm 1. A representative sample of traces $L' \subseteq L$ should reflect the characteristics that allow describing the complete event log, that is, $L'$ should allow inferring the behavior of $L$. Therefore, if a biased $L'$ is obtained, its usefulness is limited, depending on the degree of bias in the set. For the traces in $L'$ to be representative, these traces must be obtained from an appropriate sampling or instances selection technique. Commonly, instance selection techniques have as a goal to remove redundant and noisy instances from a given data set, while keeping or possibly improving performance than the original data set. In other words, instance selection techniques enable to obtaining a data set of manageable reduced size, thus reducing the computational

resources to process such data. The $L'$ size is expected to be much lower than that of $L$, although large enough for having an adequate confidence level for the computed conformance metric.

$$nSample = \frac{Z_\alpha^2 npq}{e^2(n-1) + Z_\alpha^2 pq} \tag{2}$$

From *Martínez Bencardino (2012)*, the statistical measure of the representative sample size presented in Eq. (2) was used, where $n$ corresponds to $|L|$, $Z_\alpha^2$ is a constant that depends on the confidence level. The confidence level indicates the probability that the results are true. $Z_\alpha^2$ is obtained from the standard normal distribution table. The higher the confidence level, the larger the sample size should be. The parameter $e$ corresponds to the sampling error between the sample and the complete event log. $p$ is the portion of elements in the event log that possess the expected behavior of the complete set of traces, *i.e.*, the portion of relevant instances in the log, and $q$ is the proportion of elements that do not have that behavior. Generally, $p$ and $q$ are unknown and therefore assumed as $p = q = 0.5$.

The statistical measure in Eq. (2) serves as a reference value to establish the approximate amount of traces (*nSample* at line 22 in Algorithm 1) that should be considered in the instance selection task so that a better description of the behavior observed in the event log can be achieved. Also, in this same stage, a probabilistic selection technique of instances was used to randomly select a small $L'$ from $L$. For this, simple random sampling was used, which eliminates the bias in the selection of traces since all traces in $L$ have the same opportunity to be part of $L'$.

At line 23 of Algorithm 1, the random sample of traces for Stage 3 is computed by calling the `randomSampling` function, implemented by Algorithm 2. The function `getSimilarity`$(t_1, t_2)$ implemented by Algorithm 3 returns the degree of similarity between two selected traces in $L$. At lines 4 and 7 in Algorithm 2, traces are selected uniformly at random and included in $L'$ only if both are not already in the sample subset under construction and if their similarity is less than $\gamma$ (no similar traces). Otherwise, the traces are discarded. This similarity between traces is only assessed at each iteration ($i$-th iteration). The process of building a set of representative instances from the event log is carried out until the size of the representative sample *nSample* is reached.

The classic similarity metric (*Levenshtein, 1966*) was used for the similarity calculation. Levenshtein distance is a metric that determines the minimum number of edits (insertions, removes or substitutions) that are necessary to change from one string to another. In our context, this metric determines the minimum number of editions to transform one trace into another. In Algorithm 3 (lines 1 and 2), the length of the two traces is first obtained. If one of the traces is empty, the algorithm returns zero as the similarity value. If not, the algorithm determines the necessary transformations (insertions, deletions, or substitutions) to convert one trace to another. In line 7, a matrix $T[m+1][n+1]$ is defined, being $n$ and $m$ the lengths of the two traces. This matrix is used to store the distance of the transformations to be performed from one trace to another. If the same event exists at the same position of the trace, its cost is 0 (line 16). Otherwise, it costs 1.

---

**Algorithm 2** Random sampling algorithm.

---

**Data:** $L$: event log; $nSample$: sample size;

**Result:** $L'$: the sample traces subset

1  $L' \leftarrow \varnothing$;

2  **repeat**

3      **repeat**

4          $t_i \leftarrow randomSelect(L)$;

5      **until** $t_i \notin L'$;

6      **repeat**

7          $t_j \leftarrow randomSelect(L)$;

8      **until** $t_j \notin L'$;

9      $sim \leftarrow getSimilarity(t_i, t_j)$;

10      **if**$(sim < \gamma)$ **then**

11          $L' \leftarrow L' \cup \{t_i, t_j\}$;

12      **end**

13  **until** $(|L'| < nSample)$;

---

---

**Algorithm 3** `getSimilarity` $(t_i, t_j)$ **function for computation of traces similarity.**

---

**Data:** $t_i, t_j$: traces

**Result:** $sim$: similarity value of input traces

1  $m \leftarrow t_i.length$;

2  $n \leftarrow t_j.length$;

3  **if** $(m == 0 \; OR \; n == 0)$ **then**

4      $sim \leftarrow 0$;

5      **return** $sim$

6  **else**

7      $T[m+1][n+1] \leftarrow \{0\}$;

8      **for** $k = 1 \in m$ **do**

9          $T[k][0] \leftarrow k$

10      **end**

11      **for** $k = 1 \in n$ **do**

12          $T[0][k] \leftarrow k$;

13      **end**

14      $cost \leftarrow 0$;

15      **for** $k = 1 \in m$ **do**

16          **for** $l = 1 \in n$ **do**

17              **if** $(t_i[k-1] == t_j[l-1])$ **then**

---

| Algorithm 3 (continued) | |
|---|---|
| 18 | $cost \leftarrow 0$; |
| 19 | **else** |
| 20 | $cost \leftarrow 1$; |
| 21 | **end** |
| 22 | $T[k][l] \leftarrow minimum(T[k-1][l]+1, T[k][l-1]+1, T[k-1][l-1]+cost)$; |
| 23 | **end** |
| 24 | **end** |
| 25 | **end** |
| 26 | $distance \leftarrow T[m][n]$; |
| 27 | $sim \leftarrow 1 - \frac{distance}{max(m,n)}$; |
| 28 | **return** $sim$ |

The minimum cost is obtained from the distance matrix, being any of the three transformations (line 22).

The input trace obtained by random sampling is valid. In an environment with a large amount of traces and low dispersion in the event log, the random selection of traces allows to reduce the bias in the sample. Thus, the behavior of the complete event log can be recovered from the subset of traces resulting from the random sampling. However, if the level of dispersion is high, the randomly selected subset of traces may not always represent the behavior of the full log and therefore the conformance checking task of the process model may fail.

***Using the full event log (no sampling)***

If $dispDP > \alpha$, all the traces in $L$ are used as $L'$ for the conformance checking task in stage 3 (line 27 in Algorithm 1). In this case, the dispersion level is high, indicating that extracting a trace subset would inadequately represent the complete log's behavior. This also applies when $|L| \leq minTraces$. Therefore, it's recommended to sample representative traces when $dispDP \leq \alpha$, as that subset can sufficiently capture the complete event log behavior.

## Conformance metric

Stage 3 of the proposed method uses a fitness function to calculate the conformance value between the process model $M$ and the representative sample of traces $L'$. The fitness function is shown in Eq. (3). It calculates the percentage of the sample's behavior replayable by the process model, *i.e.*, it determines the degree of correspondence or flow of control between a process model and its trace sample. Each time a trace in the event log achieves optimal alignment with the model, the alignment cost is reduced so that the set of traces contributes to determining the full alignment cost. A fitness value equal to 1 indicates that all the behaviors in the event log are described by the process model, or in other words, all traces in the event log are replayed in the process model.

$$fitness(L, M) = 1 - \frac{fcost(L, M)}{move_L(L) + |L| \times move_M(M)}. \tag{3}$$

In Eq. (3), $fcost(L, M)$ is a standard cost function as described in "Conformance Checking", and $move_L$ and $move_M$ correspond to the number of moves in the log and model, respectively.

## EXPERIMENTAL EVALUATION AND RESULTS

In this section, we present the results of the experimental evaluation of the proposed conformance approximation method described in the previous section, in terms of efficiency and performance. In the experiments, the response time of each of the stages of the proposed method was measured using real-life event logs. We also obtained the gain in performance due to the use of a reduced subset of the event log for the conformance metric computation.

### Data

The event logs used for experimentation are: Sepsis Cases (*Mannhardt, 2016*), regarding the sepsis medical condition of a hospital; Hospital Billing (*Mannhardt, 2017*), containing billing data for medical services; Hospital log (or BPIC2011) (*van Dongen, 2011*), with cases of patients of gynecology department; Road (*de Leoni & Mannhardt, 2015*), a log of an information system managing road traffic fines; and BPIC2012 (*van Dongen, 2012*), related to loan or overdraft applications at a Dutch financial institute. While the first three event logs are representative in the healthcare domain reported in the literature, the two last are considered for a qualitative comparison against related works. Although the datasets considered for experimentation are of different domains, all of them are suceptible to present different levels of dispersion in the execution of their activities. Besides, the datasets are event logs with a large number of cases, suitable for assessing our proposal under conditions that favor trace selection. The description of each log is presented in Table 2.

### Setup

The parameters selected for the dispersion threshold $\alpha$ and minimum number of traces *minTraces* used in Algorithm 1 were recommended according to the following criteria. The dispersion level *dispDP* is in the range [0, 1], where a value close to 0 indicates that the activities are distributed across the event log as expected. A value close to 1 indicates the opposite (activities distributed in the event log unexpectedly). Therefore, we consider that an event log has a high dispersion if *dispDP* is close to 1, medium dispersion if *dispDP* in the range [0.5, 0.7], and low dispersion if *dispDP* is below 0.5. In our conformance approximation method, it was recommended to apply the random probability sampling strategy on those event logs with a lower and medium level of dispersion, that is, $\alpha \leq 0.7$.

It is known that there is no exact way to classify event logs (or databases) into large, medium, or small size. However, since the task of conformance checking requires a high

**Table 2 Characteristics of the event logs used to evaluate and compare our proposed method.**

| Event log | Events | Cases | Activities | Trace length | Size (MB) |
|---|---|---|---|---|---|
| Sepsis Cases | 15,214 | 1,050 | 16 | 14.48 | 5.4 |
| Hospital Billing | 451,359 | 100,000 | 18 | 4.51 | 174.3 |
| BPIC2011 | 150,291 | 1,143 | 624 | 131.48 | 251.911 |
| Road | 561,470 | 150,370 | 11 | 3.7 | 3.3 |
| BPIC2012 | 262,200 | 13,087 | 23 | 20.03 | 3.2 |

computational cost when working with large logs, the dispersion level is only computed in those logs of medium to large size, that in this work are those logs with more than 100 traces. Othercase, small logs (with less than 100 traces) are no considered in our conformance checking strategy.

## Experiment 1. Calculation of the event log dispersion level

In the first part of the experimental evaluation, a study of the dispersion level of the event log data was carried out considering the three logs described in the previous section Data.

Table 3 presents in detail the dispersion level obtained for each activity recorded in the event log. The dispersion level behaves well for the activities with the highest frequencies, obtaining a low dispersion level. Column 2 lists the activities in the event log; column 3 presents the dispersion level for each activity; column 4 the occurrence frequency of activity in the event log; and finally, column 5 presents the dispersion type. Due to the large number of activities in the BPIC2011 event log, only 10 of these activities are presented.

Also, Table 4 shows the average dispersion level obtained by the three event logs, where BPIC2011 presents the highest level, possibly due to the wide variety of activities and the diversity in the size of its traces. The three logs present high and low dispersion levels according to the previously recommended classification; thus, the sample of traces was computed in all cases. Also, Table 4 shows the size of the trace sample obtained according to Eq. (2). In all cases, the subset of traces for the conformance checking task was much less the size of the entire event log: 80%.

To understand the practical application of the dispersion level in the sampling process, we conducted an analysis to explore its correlation with performance metrics, specifically, the accuracy loss in fitness and the increase in processing time. To achieve this, we constructed two synthesized event logs: one with 110 traces (Log110) and another with 200 traces (Log200). We intentionally introduced varying degrees of dispersion among the activities within each trace in both logs. This allowed us to investigate whether a correlation exists between the dispersion level and both fitness accuracy and processing time. Table 5 shows the results obtained from the experiment conducted. From the presented results, it is concluded that whenever the level of dispersion within the event log rises, both fitness accuracy and execution time are impacted. While the effect may occasionally be marginal, there exists a correlation between the dispersion metric and the accuracy of fitness as well as processing time.

**Table 3 Dispersion level of activities for each event log.**

| Event log | Activity | DP | Frequency | Dispersion type |
|---|---|---|---|---|
| Sepsis cases | ER-Registration | 0.217 | 1,050 | Low |
| | Leucocytes | 0.161 | 3,383 | Low |
| | CRP | 0.143 | 3,262 | Low |
| | LacticAcid | 0.260 | 1,466 | Low |
| | ER triage | 0.200 | 1,053 | Low |
| | ER sepsis triage | 0.212 | 1,049 | Low |
| | IV Liquid | 0.284 | 753 | Low |
| | IV Antibiotics | 0.243 | 823 | Low |
| | Admission NC | 0.253 | 1,182 | Low |
| | Release A | 0.332 | 671 | Low |
| | Return ER | 0.654 | 294 | Medium |
| | Admission IC | 0.771 | 117 | High |
| | Release B | 0.933 | 56 | High |
| | Release C | 0.963 | 25 | High |
| | Release D | 0.965 | 24 | High |
| | Release E | 0.994 | 6 | High |
| Hospital billing | New | 0.005 | 101,289 | Low |
| | Fin | 0.005 | 74,738 | Low |
| | Release | 0.004 | 70,926 | Low |
| | Code Ok | 0.005 | 68,006 | Low |
| | Billed | 0.004 | 67,448 | Low |
| | Change Diagn | 0.014 | 45,451 | Low |
| | Delete | 0.413 | 8,225 | Low |
| | Reopen | 0.468 | 4,669 | Low |
| | Storno | 0.465 | 2,973 | Low |
| | Set status | 0.496 | 705 | Low |
| | Empty | 0.496 | 449 | Low |
| | Reject | 0.505 | 2,016 | Medium |
| | Code Nok | 0.539 | 3,620 | Medium |
| | Change end | 0.501 | 38 | Medium |
| | Manual | 0.503 | 372 | Medium |
| | Join-pat | 0.503 | 358 | Medium |
| | Code error | 0.500 | 75 | Medium |
| | ZDBC BEHAN | 0.502 | 1 | Medium |
| BPIC2011 | Echografie-genitalia interna | 0.815 | 249 | High |
| | Kaart kosten-out | 0.720 | 218 | High |
| | Simulator-gebruik voor aanvang megavol | 0.796 | 181 | High |
| | Trombotest | 0.947 | 51 | High |
| | Brachytherapie-interstitieel | 0.728 | 153 | High |
| | Inwend geneesk | 0.8426 | 61 | High |
| | Natrium-vlamfotometrisch | 0.933 | 24 | High |
| | Vitamine b12 mbv chemieluminescentie | 0.961 | 14 | High |
| | Bloedgroep abo en rhesusfactor | 0.950 | 28 | High |
| | Trombocyten tellen-spoed | 0.945 | 26 | High |

**Table 4 Dispersion level and sample size of event logs.**

| Log features | Event logs | | |
|---|---|---|---|
| | Sepsis cases | Hospital billing | BPIC2011 |
| Dispersion level | 0.473 | 0.409 | 0.907 |
| Traces | 1,050 | 100,000 | 1,143 |
| Sampled log size | 282 | 383 | 288 |

**Table 5 Correlation between the dispersion level, fitness, and execution time over two synthesized event logs of 110 traces (Log110) and 200 traces (Log200).**

| Event log | Dispersion level | Fitness | Runtime (ms) |
|---|---|---|---|
| Log110 | 0.781 | 0.911 | 47.6 |
| Log110 | 0.665 | 0.911 | 43.5 |
| Log110 | 0.396 | 0.933 | 31.8 |
| Log200 | 0.801 | 0.894 | 57.6 |
| Log200 | 0.526 | 0.890 | 55.6 |
| Log200 | 0.337 | 0.862 | 52.3 |

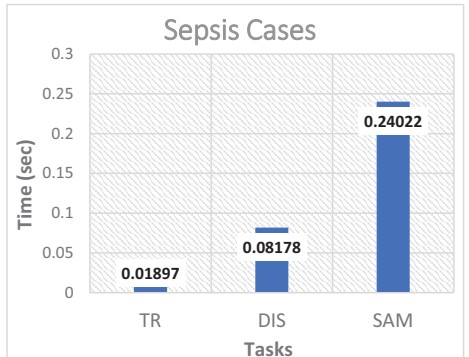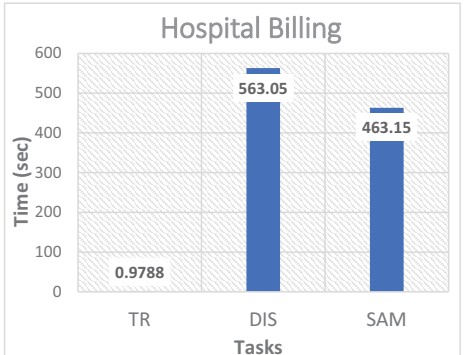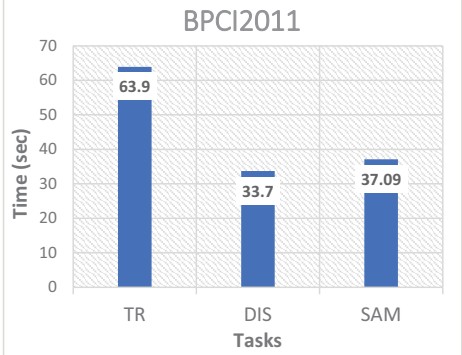

**Figure 4 Processing time from the transformation of the event log to the conformation of the subset of trace sample.**

## Experiment 2: performance evaluation

Figure 4 shows the performance evaluation of the proposed method, expressed as the response time that was calculated as the sum of the timing for the different main processes: event log transformation *TR*, calculation of the dispersion level *DIS*, and the calculation of the size of the representative sample *SAM*. All these processes cover the data processing cycle, starting from the transformation of the raw event log until the traces sample subset is obtained. According to the obtained results (range from 0.09 s to over 500 s), it is hard to generalize the method's behavior in the *TR*, *DIS*, and *SAM* processes. In two of the three event logs, *TR* was negligible if compared to *DIS* or *SAM*. However, that is not the case for

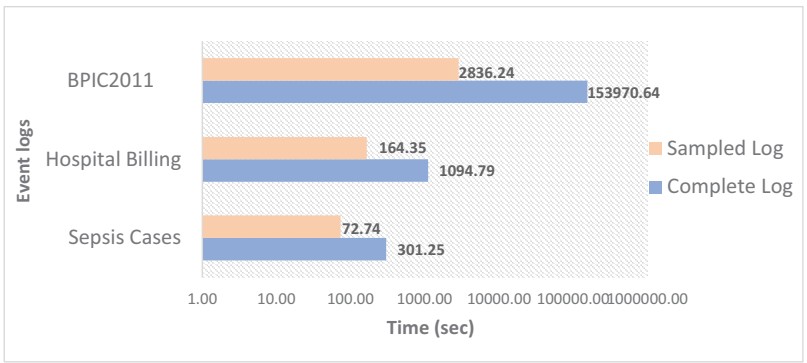

**Figure 5  Conformance checking time using the complete and sampled event logs.**

the event log of BPIC2011, where the *TR* cost is almost half the total response time due to the number of activities considered by BCPI11, as well as the average size per trace, which is up to nine times larger compared with the size of the traces in Sepsis Cases and Hospital Billing. Note that the Sepsis Cases and BPIC2011 have a similar number of traces but not similar trace length, dispersion level, and amount of activities, which translates into higher processing times in the three processes *TR*, *DIS* and *SAM*, with considerably larger distance between these two event logs. In this same test, only the response time to calculate the conformance value was obtained for both cases by using the complete event log and when using only the sampled event log.

Figure 5 shows the time for computing the conformance value for the three event logs. It can be noted that the greatest time corresponds to BPIC2011. Despite not being the log with the largest number of cases compared to Hospital Billing, the BPIC2011 log is characterized by containing very long traces, and the dispersion level is higher than the other two event logs, which makes the conformance checking process much slower compared to the other two logs. However, Fig. 5 reveals that the time is considerably less when subsets of samples from the original event logs are used. In all cases, there are notable time savings when using sampled event logs, from 75% with the Sepsis Cases sampled log to 98% with the BPCIP2011 sampled log.

## Experiment 3: measuring the conformance value

In this third experiment, the conformance value was assessed considering the alignments computed against every trace in the log as baseline. While improving the processing time of the conformance-checking task is important, achieving conformance values similar to those obtained with the original event log is essential to ensure that there is no high loss of process behavior. For this purpose, the conformance value between process models and the complete and sampled event logs was analyzed.

Table 6 shows the results obtained from conformance assessment using the fitness metric (Eq. (3)) under the two scenarios described previously. Together with fitness, we also provide in Table 6 results for precision and generalization to show that the use of a sampled trace set, selected as proposed in our article, does not greatly affect these common

**Table 6 Performance comparison obtained using the complete event log and the sampled log by proposed approach.**

| Event log | Metric | Complete | Sample | Loss |
|---|---|---|---|---|
| Sepsis cases | Fitness | 0.901 | 0.899 | 0.002 |
| | Precision | 0.503 | 0.453 | 0.050 |
| | Generalization | 0.998 | 0.992 | 0.006 |
| | F-score | 0.645 | 0.602 | 0.043 |
| Hosp. billing | Fitness | 0.919 | 0.901 | 0.018 |
| | Precision | 0.557 | 0.409 | 0.148 |
| | Generalization | 0.998 | 0.987 | 0.011 |
| | F-score | 0.694 | 0.563 | 0.131 |
| BPIC2011 | Fitness | 0.935 | 0.861 | 0.074 |
| | Precision | 0.625 | 0.485 | 0.140 |
| | Generalization | 0.986 | 0.9800 | 0.006 |
| | F-score | 0.758 | 0.625 | 0.133 |

metrics used in process mining. On the one hand, precision estimates the extent of behavior allowed by the process model that is not observed in the event log. On the other hand, generalization estimates the extent of behavior allowed by the process model that is not observed in the event log. Both, precision and generalization were obtained from the ProM tool. Columns 3 and 4 show the results of the evaluation obtained using the complete and sampled event log built according to the proposed trace-based sampling method. The last column presents the loss in each metric for the case when using the sample subset of traces instead of the complete event log. According to these results, this loss, particularly in fitness, is small in most cases. For example, for the BPIC2011 log, the loss is 7.4%. A possible cause of this situation is the level of dispersion and the number of activities considered by BPIC2011. For event logs characterized by a high level of dispersion, such as BPIC2011, it is advisable to utilize the entire set of traces rather than applying sampling to mitigate the risk of experiencing substantial fitness loss. In the case of the precision metric, the loss is most notable in both Hospital Billing and BPIC2011 logs, which results in the process model admitting behavior not seen in the event log.

Also, this third experiment compared the conformance level obtained for the three event logs under three different configurations. This is shown in Fig. 6, where the light blue bars represent the conformance value between the process model (discovered from the complete log) and the complete log (case referred to as CCL); the orange bars show the conformance value between the process model (discovered with the sampled log) and the sampled log (case referred as SSL); finally the grey bars show the conformance level between the process model (built with the complete log) and the sampled log (case referred as CSL). This is to verify if there is a loss or gain in the level of conformance obtained by reducing behavior in the process model discovered with a sampled event log. At the same time, to analyze how that difference is concerning the three configurations to determine which of these configurations has a better conformance level.

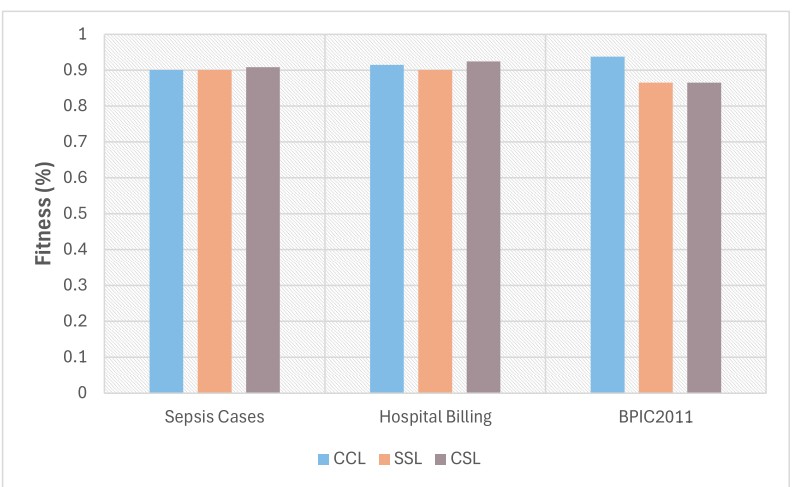

**Figure 6 Fitness using three different configurations: CCL, SSL, and CSL.**

As shown in Fig. 6, the fitness obtained for the Sepsis Cases and Hospital Billing logs in the CSL configuration is better than the one obtained in CCL or SSL. These cases demonstrate that the method for selecting a traces subset retains the process's behavior and better fits the model. In the case of the BPCI2011 log, the fitness value is maintained for both SSL and CSL configurations. However, there is a noted difference between the latter two configurations and CCL due to the particular characteristics in the event log, such as a high level of dispersion and many activities. In all cases, a small loss in fitness was experienced because the process model was obtained with a sample of the event log.

# RELATED WORK AND COMPARISONS

In recent years, various techniques have been proposed to address the conformance-checking problem by considering a trace alignment approach based on trace sampling and approximation, as in our proposal. *Bauer, van der Aa & Weidlich (2022)* presented a statistical approach to conformance checking that employs trace sampling and result approximation to derive conformance results. The authors described mechanisms to reveal biases in sampling procedures and proposed methods to assess the internal and external quality of a trace sample based on behavioral representativeness and data attribute coverage. They used the Euclidean distance as a measure of similarity between two traces. In contrast to that work, we use the Levenshtein distance to determine the minimum number of edits (insertions, removes or substitutions) that are necessary to change from one trace to another. We apply this measure to determine the similarity between two traces. This metric allows us to know the minimum number of changes needed to convert one trace to another and hence conclude if they are referring to the same trace or not.

In *Bernard & Andritsos (2021)*, the problem of sampling event logs is solved by three log-sampling algorithms minimizing the Earth Movers' Distance (EMD) using a linear programming formulation. The authors presented Expected Occurrence Reduction (EOR), a heuristic to reduce the computation time and turning a variant-based sampling into a

trace-based one, allowing repetitions when the sampling technique being used does not support that feature. When the number of unique variants is less than one or two thousand, the authors suggest applying EOR and Iterative c-min (Eucl.) for larger events. One problem in that work is that when the number of unique variants exceeds a few thousand, the time and space needed to build the cost matrix grow significantly, making EOR inefficient. In contrast, our proposal considers approximating the conformance value when having a large number (exceeds a few thousand) of unique traces, without affecting the time, space and number of unique variants used in the calculation of the conformance value.

*Sani, van Zelst & van der Aalst (2019)* proposed approximation techniques to compute approximated conformance-checking values using a statistical approach. Those techniques use upper and lower bounds for the approximated alignment value and consider a subset of the process model behavior instead of all its behavior. The authors proposed three different methods to select traces in the subset, randomly or based on their frequency in the event log and the third method that consists of applying a clustering algorithm on the event log using control follow information of traces as similarity criterion. The authors showed that the frequency method is more applicable to selecting the candidate traces with accurate results. In contrast to the work of *Sani, van Zelst & van der Aalst (2019)*, we consider the development of a trace sampling approach based on the calculation of the level of dispersion of events in the log and the sample size of the event log.

The work of *Sani, van Zelst & van der Aalst (2019)* was considered as a baseline for the development of our proposal. We use a random selection method for the identification of candidate traces to be part of the subset, as well as the Levenshtein distance measure to determine the similarity between two traces. Also, we used the Inductive Miner algorithm for the discovery of process models, which allowed us to perform the conformance process with the event logs. Unlike the work of *Sani, van Zelst & van der Aalst (2019)*, we calculate the number of traces that are part of the subset of traces representative of the behavior of the complete set of event logs using a statistic approach. Also, we obtained the level of dispersion that exists in the events recorded in the log, to determine the most appropriate trace selection method.

Some other works have also addressed the trace alignment problem using other approaches, these are described below. *Bose & van der Aalst (2010)* presented a trace alignment approach inspired by progressive sequence alignment, where a succession of pairwise alignments is iteratively constructed. Alignment is done between a pair of traces. The selection of traces for alignment at each iteration is based on their similarity. Traces that are most similar to each other are aligned first. Then, the traces in the resulting clusters are aligned with each other.

*de Leoni, Maggi & van der Aalst (2012)* proposed a conformance-checking approach based on an alignment that minimizes the cost of searching for deviations using the A* algorithm. The authors provided diagnostics coloring constraints in the model based on their degree of conformance. *Rozinat & van der Aalst (2008)* proposed an incremental approach to check the conformance of a process model and its event log. Their approach

introduces the dimension appropriateness to capture the idea that "one should not increase, beyond what is necessary, the number of entities required to explain anything".

Big data has reached PM, so considerable amounts of process events are being collected and stored by organizations. Some alignment techniques (*Leemans, Fahland & van der Aalst, 2018*; *Reiner et al., 2020*; *Schuster & Kolhof, 2021*) have considered scalability to work with large event logs of billions of events or thousands of activities, incorporating parallelism strategies or additional worker nodes. Working with large volumes of process events implies no guarantee of having solid models.

*Leemans, Fahland & van der Aalst (2018)* proposed a model–model and model–log comparison framework that relies on a divide-and-conquer strategy. The framework projects the system and system model/log onto subsets of activities to determine their recall/fitness and precision. *Reiner et al. (2020)* presented an approach where the problem of conformance checking can be mapped to computing a minimal partially synchronized product between an automaton representing the event log and an automaton representing the process model. The resulting product automaton can be used to extract optimal alignments between each trace in the log and the closest corresponding trace of the model.

*Schuster & Kolhof (2021)* presented a scalable online conformance-checking approach based on incremental prefix-alignment calculation. This work is based on Apache Kafka, a streaming platform used within the industry. Prefix alignment computation is carried out on an event stream by continuing a shortest path problem based on an extended search space upon receiving a new event.

Recently, some of the approaches for alignment have been aimed at estimating the conformance value through approximation techniques. *van der Aa, Leopold & Weidlich (2021)* presented a process conformance checking approximation method based on partial order resolution of event logs to cope with the order uncertainty in real-world data. Conformance checking is grounded in a stochastic model that addresses the problem of partial order resolution of event logs by incorporating a probability distribution over all possible total orders of events of an instance. In *Sani et al. (2020)*, the authors used simulated behaviors of process models to approximate the conformance-checking value. These simulated behaviors are close to the recorded behaviors in the event log. Also, the authors used the edit distance function and provided bounds for the actual conformance value. The method additionally returns problematic activities based on their deviation rates.

According to the previous section, our proposal is strongly related and shares similar objectives to three works: *Sani, van Zelst & van der Aalst (2019)*, *Bauer, van der Aa & Weidlich (2022)*, and *Bernard & Andritsos (2021)*. Table 7 presents the results obtained by the four different approaches (the ours included) using the same event logs and tool (Inductive Miner). The metric *PI* (used in *Sani, van Zelst & van der Aalst (2019)*) is used to measure how the conformance approximation can improve the performance of the conformance checking process. *PI* is obtained as the Normal Conformance Time divided by Approximated Conformance Time. This metric is presented as intervals corresponding to the overall performance obtained with different approximation parameter. It can be observed in Table 7 that our proposal shares fitness values very close to those of the other

**Table 7 Quantitative comparison of related works using an approximation conformance approach.**

| Approach | Event log | Fitness | PI | Run time |
|---|---|---|---|---|
| *Sani, van Zelst & van der Aalst (2019)* | Sepsis cases | 0.926 | $[5 - 10^2]$ | 120.3 s |
| *Bauer, van der Aa & Weidlich (2022)* | | 0.887 | $[10^1 - 10^2]$ | 98.1 s |
| *Bernard & Andritsos (2021)* | | 0.901 | $[0.1 - 6 \times 10^2]$ | 85.2 s |
| Our proposal | | 0.899 | $[6 - 10^1]$ | 73.0 s |
| *Sani, van Zelst & van der Aalst (2019)* | Hospital billing | 0.770 | $[10^1 - 10^2]$ | 368.5 s |
| *Bauer, van der Aa & Weidlich (2022)* | | 0.925 | $[5 - 10^2]$ | 140.2 s |
| *Bernard & Andritsos (2021)* | | 0.899 | $[5 - 10^2]$ | 336.4 s |
| Our proposal | | 0.901 | $[4 - 10^1]$ | 164.1 s |
| *Sani, van Zelst & van der Aalst (2019)* | Road | 0.822 | $[10^0 - 10^1]$ | 76.5 s |
| *Bauer, van der Aa & Weidlich (2022)* | | 0.992 | $[10^0 - 10^1]$ | 61.5 s |
| *Bernard & Andritsos (2021)* | | 0.876 | $[6 - 10^1]$ | 38.3 s |
| Our proposal | | 0.899 | $[5 - 10^1]$ | 55.2 s |
| *Sani, van Zelst & van der Aalst (2019)* | BPIC-2012 | 0.814 | $[10^1 - 10^2]$ | 1,045 |
| *Bauer, van der Aa & Weidlich (2022)* | | 0.948 | $[7 - 10^2]$ | 730 |
| *Bernard & Andritsos (2021)* | | 0.814 | $[10^1 - 10^2]$ | 550 |
| Our proposal | | 0.902 | $[10^1 - 10^2]$ | 685 |

works in most of the cases. Although the fitness obtained by our proposal is not superior with the different event logs. The use of sampling manages to decrease the runtime in most of the cases.

Table 8 presents a qualitative comparison. Column 2 shows the methods developed by the three works and our proposal for the selection of the trace sample that represents the behavior of the complete log. The next column presents the distance metrics used to measure the similarity between two traces. In column 4, *A* corresponds to the use or lack of an approximation approach of the conformance value; *B* corresponds to the use of upper-lower bounds to determine the conformance value; *S* refers to the use of some metric to calculate the number of traces that make up the subset of the representative sample of traces; *D* corresponds to the use of a metric of the level of dispersion of the events in the log. The *Dis* column refers to the process model discovery algorithm used by each of the articles. Finally the metric *Acc* (used in *Sani, van Zelst & van der Aalst (2019)*) presented in the last column is used to measure how close the approximated fitness to the actual fitness value is, respectively. *Acc* is obtained as |NormalFitness - AppxFitness|.

The results shown in *Acc* are presented as intervals corresponding to the overall performance obtained with different event logs. In this same Table 8, it can be seen that the related works and our proposal share similar times and, in some cases, our proposal manages to improve the time obtained in *PI*. The difference between the real and approximate fitness (Acc) is relatively small in most cases.

*Bauer, van der Aa & Weidlich (2022)* claimed that their approach can improve the runtime of conformance checking, without sacrificing the accuracy of the overall

**Table 8 Comparison of works for conformance checking using a sampling approach.**

| Works | Methods | Distance | A | B | S | D | Dis | Acc |
|---|---|---|---|---|---|---|---|---|
| *Sani, van Zelst & van der Aalst (2019)* | Randomly frequency clustering | Levenshtein | ✓ | ✓ | X | X | Inductive | [0.003 − 0.11] |
| *Bauer, van der Aa & Weidlich (2022)* | Statistical | Euclidean | ✓ | ✓ | X | X | Inductive | [0.00049–0.00223] |
| *Bernard & Andritsos (2021)* | Iterative c-m EOR Eucl. | Earth Mov | ✓ | X | X | X | – | |
| Our proposal | Randomly sample size | Levenshtein | ✓ | X | ✓ | ✓ | Inductive | [0.002–0.018] |

**Table 9 Generalities of conformance checking approaches in the literature for different scenarios.**

| Ref. | Approach | Modeling language | | | | Perspective | Quality measures | | |
|---|---|---|---|---|---|---|---|---|---|
| | | BPMN | PN | Declare | DFG | | F[1] | P[2] | G[3] |
| *Rozinat & van der Aalst (2008)* | Incremental | | ✓ | | | Control-flow | ✓ | ✓ | ✓ |
| *Bose & van der Aalst (2010)* | Trace alignment | | ✓ | | | Control-flow | | | |
| *de Leoni, Maggi & van der Aalst (2012)* | Trace alignment | | | ✓ | | Control-flow | ✓ | | |
| *Lu, Fahland & van der Aalst (2015)* | Partially ordered traces and alignments | | ✓ | | | Control-flow | | | |
| *Song et al. (2016)* | Heuristics and trace relaying | | ✓ | | | Control-flow | ✓ | | |
| *de Leoni & Marrella (2017)* | Alignment as a planning problem | | ✓ | | | Multiple alignments | | | |
| *Burattin et al. (2018)* | Online, behavioural patterns | | ✓ | | | Control-flow | ✓ | | |
| *Leemans, Fahland & van der Aalst (2018)* | Scalable, divide and conquer strategy | | ✓ | | | Control-flow | ✓ | ✓ | |
| *Sani, van Zelst & van der Aalst (2019)* | Approximation, upper and lower bounds | | ✓ | | | Multiple alignments | ✓ | | |
| *Polyvyanyy & Kalenkova (2019)* | Partial matching between traces | ✓ | ✓ | | | | ✓ | ✓ | |
| *Sani et al. (2020)* | Approximation, simulated behaviors | | ✓ | | | Control-flow | ✓ | | |
| *van der Aa, Leopold & Weidlich (2021)* | Approximation, partial order | ✓ | | | | Control-flow | ✓ | | |
| *Felli et al. (2022)* | Satisfiability Modulo Theories (SMT) encoding | | ✓ | | | Optimal alignment | | | ✓ |
| *Schuster et al. (2022)* | Infix/postfix alignment computation | | ✓ | | | Control-flow | | | |
| *Rocha & van der Aalst (2023)* | Markovian abstraction | | ✓ | | | Multiple alignments | ✓ | ✓ | |
| *Park, Adams & van der Aalst (2024)* | Object-Centric Directly Follows Graphs (OC-DFGs) | | ✓ | | ✓ | Control-flow | | | |

conformance results. According to the reported results, the sample sizes are from 0.5% to 10% of the size of the original logs. In contrast to our proposal, we obtain sample sizes from 0.38% to about 25% of the original log size. For the logs where the level of dispersion of events is high, the suggested size to form the subset of traces is around 25%. This is done to ensure a representative sample of the complete behavior of the original log. Also, Bauer's approach achieves fitness results from 5% to 51.2% of the time required for different logs. In comparison to our approach, we were able to achieve the fitness calculation from 1.8%

to 24% of the time required if the event log was processed with the sampling technique we proposed. This indicates that also our approach achieves high performance in the runtime of conformance checking.

Table 9 shows the generalities of alignment techniques described in this section. Most of these techniques use Petri nets (PNs) for process modeling in the alignment task. PNs are one of the most commonly used models for PM strategies as they provide a clear view of the dependencies between process activities and allow for easy monitoring of the process. About half of the alignment techniques presented in this table use approaches based on conformance or partial conformance approximation. In addition, this table shows that most alignment techniques focus on the control-flow perspective, analyzing the order of steps or activities to determine the conformity between prescribed and actual behavior. Few of the alignment works use various quality measures to evaluate their performance.

According to the related works described so far, different approaches based on the alignment technique can be distinguished. Some works are focused on improving the structural behavior of the process model by presenting it more simply, without admitting many process behaviors to help the alignment task. Other works incorporate activity parallelization techniques to accomplish the alignment task. Some other works consider clusters of traces so that the alignment is performed on a pair of traces from each cluster without considering the complete event log. This idea of reducing the amount of traces in the event log for the conformance task can improve the performance of the alignment technique. However, it is worth noting that for this purpose, it is essential to know the adequate number of instances to consider as the representative sample of traces with the least negative impact in the conformance metric.

## CONCLUSIONS

A novel method for the conformance-checking task in process mining was discussed. It uses an instance selection approach based on the data dispersion in the event log. Then, an approximation algorithm is used for conformance calculation. Thus, our proposed method finds a value close to the exact conformity solution but only considering a representative traces samples subset of the event log. The main advantage of this approach is a considerable reduction of computational workload with the most negligible impact on fitness. The instance selection mechanism obtains a representative number of traces in the event log using a statistical measure as an approximate but effective traces set for conformance computation. The novel conformance method reported in this work considers two important elements: on the one hand, the calculation of the level of dispersion of the event log that helps to determine whether it is necessary or not to make the selection of representative instances; and the definition of the sample size that allows knowing the most appropriate size of data to be considered as part of the sample.

The novel method significantly reduces the computational cost (up to 90%) without a significant loss of conformance value if it was obtained using the complete event log. Therefore, the proposed method is suitable for a Big Data environment where massive data is available and it is highly desirable to reduce the input traces set to the conformance task.

The method's practicality was demonstrated from the experimental evaluation to asses the execution times and performance in the conformance task.

Based on the findings presented in this work, the loss incurred by employing sampled event logs for conformance verification tasks was generally minimal, with variances falling below 7.5%. However, greater differences were observed in the precision metric, indicating that the process model could accommodate behaviors not observed in the original event log.

At the moment, the method has been tested with three individual event logs from a single organization in the health domain and two artificial event logs. As part of future work, we expect to extend our approach to an inter-organizational process environment, where other challenges, such as integration and instances selection from new data sources, are considered. Furthermore, algorithms for real-time trace selection are required under such scenarios, that address the changing process behaviors and ensure that the most relevant and representative instances are consistently used.

### Funding
The authors received no funding for this work.

### Competing Interests
The authors declare that they have no competing interests.

### Author Contributions
- Heidy M. Marin-Castro conceived and designed the experiments, performed the experiments, analyzed the data, performed the computation work, authored or reviewed drafts of the article, and approved the final draft.
- Miguel Morales-Sandoval conceived and designed the experiments, performed the experiments, analyzed the data, authored or reviewed drafts of the article, and approved the final draft.
- José Luis González-Compean performed the computation work, prepared figures and/or tables, authored or reviewed drafts of the article, and approved the final draft.
- Julio Hernandez analyzed the data, performed the computation work, prepared figures and/or tables, authored or reviewed drafts of the article, and approved the final draft.

### Data Availability
The raw data are available in the Supplemental Files.

### Supplemental Information
Supplemental information for this article can be found online at http://dx.doi.org/10.7717/peerj-cs.2601#supplemental-information.

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
