# Peer review of "A novel trace-based sampling method for conformance checking"

_PeerJ Computer Science, doi:10.7717/peerj-cs.2601_

## Round 0.1 · original submission · Minor Revisions

Dear authors,

Thank you for submitting your Literature Review article. Feedback from the reviewers is now available. We strongly recommend that you address the issues raised by the reviewers and resubmit your paper after making the necessary changes. Reviewer1 and Reviewer1 have suggested you could consider specific references. You are welcome to add them if you think they are relevant and useful . However, you are under no obligation to include them, and if you do not, it will not affect my decision.

Best wishes,

·

Basic reporting

1. Introduction section should include more references about PM, such as the following:
Sypsas, A., & Kalles, D. (2022). Reviewing process mining applications and techniques in education. Int. J. Artif. Intell. Appl, 13, 83-102.
A. Polyvyanyy, A. Moffat, and L. García-Bañuelos, “An entropic relevance measure for stochastic conformance checking in process mining,” in ICPM. IEEE, 2020, pp. 97–104.
Benevento, E. et al. (2023). Process Modeling and Conformance Checking in Healthcare: A COVID-19 Case Study. In: Montali, M., Senderovich, A., Weidlich, M. (eds) Process Mining Workshops. ICPM 2022. Lecture Notes in Business Information Processing, vol 468. Springer, Cham. https://doi.org/10.1007/978-3-031-27815-0_23
2. Please explain how the given threshold (minTraces) is selected.
3. Page 11 line 359 section 28 is stated. Which section is this? Please explain.
4. Page 16 line 501 Reiner et al. Reiner et al shoiuld be corrected.
5. Page 16 line 512 van der Aalst et al. van der Aa et al. shoiuld be corrected.
6. In table 6 you could add Sypsas A, Kalles D. Analysis, Evaluation and Reusability of Virtual Laboratory Software Based on Conceptual Modeling and Conformance Checking. Mathematics. 2023; 11(9):2153. https://doi.org/10.3390/math11092153

Experimental design

Please explain in deatail how the given threshold (minTraces) is selected.
Please explain why random sampling (page 8, line 267) is valid as input trace, since instance selection techniques have as a goal to remove redundant and noisy instances from a given data set.

Validity of the findings

Conclusions could be stated in an more clear way, giving more details on the methods that were used.
Limitations could be referenced in a spare paragraph.

Cite this review as

Reviewer 2 ·

Basic reporting

no comment

Experimental design

no comment

Validity of the findings

no comment

Additional comments

This paper addresses the growing importance of process mining in data science, specifically focusing on the conformance checking task. Overall the work is good, some minor concern, that need to resolve are as follows:
1. The motivation of work should be more clear in abstract.
2. Need clarity on how dispersion level impacts trace selection.
3. The author further elabore on fitness, precision, and generalization metrics needed.
4. Explain more about figure 4.
5. Table 9, add 2022, 2023, 2024 references.
6. Better to add some lamination and future work in conclusion.

Cite this review as

Reviewer 3 ·

Basic reporting

1. The article used a very clear and unambiguous English and academic language.
2. Insufficient and irrelevant references have been provide. Some more relevant an recent references should be provided. Specifically related to Trace concept in process mining. Such as:

Grigore, I. M., Tavares, G. M., Silva, M. C. D., Ceravolo, P., & Barbon Junior, S. (2024). Automated Trace Clustering Pipeline Synthesis in Process Mining. Information, 15(4), 241.

Imran, M., Akmar Ismail, M., Hamid, S., & Hairul Nizam Md Nasir, . M. (2023). A TRACE CLUSTERING FRAMEWORK FOR IMPROVING THE BEHAVIORAL AND STRUCTURAL QUALITY OF PROCESS MODELS IN PROCESS MINING . Malaysian Journal of Computer Science, 36(3), 223–241. https://doi.org/10.22452/mjcs.vol36no3.2

Graziosi, R., Ronzani, M., Buliga, A., Di Francescomarino, C., Folino, F., Ghidini, C., ... & Pontieri, L. (2024, October). Generating the Traces You Need: A Conditional Generative Model for Process Mining Data. In 2024 6th International Conference on Process Mining (ICPM) (pp. 25-32). IEEE.

These are suggested recent reference to be added. However the author should also make sure to add more such references from recent years 2023, 2024.

3. The structure is consistent and clear presentation of figures has been used.

Experimental design

The article is within the scope of PeerJ and research questions have been well-defined, logical an sufficient reasoning have been provided.

However the reason behind using specific datasets have not be provided such as why BPIC 2011 and BPIC 2015 datasets have been used. Why not others, The explanation will be very helpful for the readers of this research to contextualize the experiment an findings.

Validity of the findings

The finding are valid and clearly reported in tubular and graphical style. Moreover, the context of the findings have also been provided.

Additional comments

The article is clear and consistent an very well written.

Cite this review as

---

## Round 0.2 · accepted · Accept

Dear Authors,

Thank you for addressing the reviewers' comments. Your manuscript now seems ready for publication.

Best regards,

·

Basic reporting

The suggestions from reviewers have been implemented.

Experimental design

Research questions and methods are explained.

Validity of the findings

Data used are presented in detail. Conclusions are well stated.

Cite this review as

Reviewer 3 ·

Basic reporting

The authors have addressed all reporting concerns as indicated in previous round of revisions.

Experimental design

The authors have addressed all experimental design concerns as indicated in previous round of revisions.

Validity of the findings

The findings are valid and inline with the comments as made in previous round of revision.

Additional comments

The article is very well written, within the scope of PeerJ and targets process mining domain. The proposed technique makes it a significant contribution to process mining research.

Cite this review as